# Efficient Removal of Levofloxacin by Activated Persulfate with Magnetic CuFe$_2$O$_4$/MMT-k10 Nanocomposite: Characterization, Response Surface Methodology, and Degradation Mechanism

**Junying Yang [1], Minye Huang [1], Shengsen Wang [2,3], Xiaoyun Mao [1,3,*], Yueming Hu [1] and Xian Chen [1,*]**

[1] College of Natural Resources and Environment, South China Agricultural University, Guangzhou 510642, China; docjunyingyang@163.com (J.Y.); minye_huang@163.com (M.H.); ymhu@scau.edu.cn (Y.H.)

[2] College of Environmental Science and Engineering, Yangzhou University, Yangzhou 225127, China; wangss@yzu.edu.cn

[3] Guangdong Provincial Key Laboratory of Eco-Circular Agriculture, South China Agricultural University, Guangzhou 510642, China

* Correspondence: xymao@scau.edu.cn (X.M.); xianchen302@scau.edu.cn (X.C.); Tel.: +86-135-8031-0522 (X.M.); Fax: +86-20-85287672 (X.M.)

**Abstract:** In this study, a magnetic copper ferrite/montmorillonite-k10 nanocomposite (CuFe$_2$O$_4$/MMT-k10) was successfully fabricated by a simple sol-gel combustion method and was characterised by X-ray diffraction (XRD), scanning electron microscopy (SEM), transmission electron microscopy (TEM), the Brunner–Emmett–Teller (BET) method, vibrating sample magnetometer (VSM), and X-ray photoelectron spectroscopy (XPS). For levofloxacin (LVF) degradation, CuFe$_2$O$_4$/MMT-k10 was utilized to activate persulfate (PS). Due to the relative high adsorption capacity of CuFe$_2$O$_4$/MMT-k10, the adsorption feature was considered an enhancement of LVF degradation. In addition, the response surface methodology (RSM) model was established with the parameters of pH, temperature, PS dosage, and CuFe$_2$O$_4$/MMT-k10 dosage as the independent variables to obtain the optimal response for LVF degradation. In cycle experiments, we identified the good stability and reusability of CuFe$_2$O$_4$/MMT-k10. We proposed a potential mechanism of CuFe$_2$O$_4$/MMT-k10 activating PS through free radical quenching tests and XPS analysis. These results reveal that CuFe$_2$O$_4$/MMT-k10 nanocomposite could activate the persulfate, which is an efficient technique for LVF degradation in water.

**Keywords:** magnetic; copper ferrite; montmorillonite-k10; levofloxacin; persulfate; response surface methodology

## 1. Introduction

In recent years, pharmaceuticals and personal care products (PPCPs), as emerging contaminants, have been detected frequently in the environment, which has caused widespread concern. Antibiotics, as one kind of PPCPs, are widely used in the treatment of pathogenic-organism diseases of humans or livestock [1,2]. However, most antibiotics are not completely transformed or metabolized after administration, which means excreta maintain the state of pharmacological activity and can be further excreted into the environment [3,4]. The long-term discharge of low-dose antibiotics into the environment will lead to the prompted drug resistance of sensitive bacteria. In addition, drug-resistant

genes might develop and interfere with environment, posing a potential threat to the ecological environmental security and human health [5]. Unfortunately, the traditional water treatment process may not meet the demands of effective removal of antibiotic contamination. It is important to develop an efficient and eco-friendly technique to remove antibiotics in waste water.

Based on free radicals, the advanced oxidation process (AOP) has been widely used in water treatment. Compared with hydroxyl radicals (HOs; $E_0$ = 1.9–2.7 eV) in the AOP [6,7], sulfate radical ($SO_4^{\cdot-}$; $E_0$ = 2.5–3.1 eV) has more advantages, such as higher redox potential, longer half-life, and wider pH range. Therefore, with regard to $SO_4^{\cdot-}$ the AOP has an advantage of pollutant removal in waste treatment. Generally, $SO_4^{\cdot-}$ is produced by peroxymonosulfate (PMS) and persulfate (PS). Compared with PMS, PS has a higher solubility, better stability, and lower cost, and is considered to be a better source of $SO_4^{\cdot-}$ [8]. Alkali [9], heat [10,11], and radiation [12,13] are good methods to activate PS and further generate $SO_4^{\cdot-}$. However, high cost and harsh activation conditions confine these approaches, as they are not practical for large-scale application. Instead, the heterogeneous catalytic oxidation technique is a potential countermeasure for these problems. Spinel ferrites ($MFe_2O_4$; M=Co, Cu, Mn, Zn, Ni) are characterized by low-cost, efficient heterogenous catalysis and easy magnetic separation from aqueous solution [14]. $CoFe_2O_4$ is considered as the most efficient catalyst for activating PS to treat organic pollutants in water [15]. However, the International Agency for Research on Cancer (IARC) has identified cobalt as a potential human carcinogen [16]. Thus, the leaching toxicity of $CoFe_2O_4$ damages its application in practical water treatment. It has been demonstrated that $CuFe_2O_4$ is more effective at activating PMS during the process of di (n-butyl) phthalate degradation than other spinel ferrites, except for $CoFe_2O_4$ [17]. Although $CuFe_2O_4$ is not as effective as $CoFe_2O_4$, copper is not recognized as potential carcinogen, according to IARC [8]. In other words, considering the catalytic activity and the risk of metal leaching of catalysts, $CuFe_2O_4$ is the optimal PS activator to choose for removing LVF in this experiment. However, $CuFe_2O_4$ nanoparticles are characterized by magnetic properties, and are likely to agglomerate together to form larger clusters, and therefore decrease the catalytic efficiency [18]. Such problems might be mostly solvable with the introduction of supporting materials.

Montmorillonite (MMT) is an abundant, typical aluminosilicate mineral with a structure of 2:1 layers consisting of one $Al^{3+}$ octahedron sheet placed between two $Si^{4+}$ tetrahedral sheets [19]. Due to its special structure, MMT possesses a large surface area, high cation exchange capacity, and smooth expansion capacity [20]. With its favorable properties and low cost, MMT is a suitable support material for various nanoparticles purposes. In recent years, quaternary ammonium salt (QAC)-modified organic montmorillonite has caught researchers' attention, due to its outstanding features listed above. Researchers have utilized cetyltrimethyl–ammonium bromide-modified organic montmorillonite as a supporting material for nanoscale, zero-valent iron to remove tetrabromobisphenol-A and bisphenol-A [21]. Researchers have also used didodecyldimethyl–ammonium bromide-modified organic montmorillonite as an absorbent material for thiophanate-methyl removal [22]. However, QACs have been released in the environment extensively in recent years, and this causes serious toxicity to protozoa [23] and shrimp [24], and even induces drug resistance in bacteria [25]. This has caused great concern from environmental scientists and bacteria researchers [26]. As a green and friendly material, montmorillonite k10 (MMT-k10) has been extensively applied in aspects of catalysis [27]. Relative to other materials, MMT-k10 offers several advantages, such as non-corrosive properties, non-toxicity, low cost, ease of handling, and mild reaction conditions [28]. Fe(III) immobilized on MMT-k10 as a catalyst have been applied in an $H_2O_2$ Fenton-like oxidation process to effectively degrade reactive black 5 and acid red 1 dye [29,30]. In the presence of $H_2O_2$, the montmorillonite K10-Cu(II)ethylenediamine catalyst could efficiently catalyze the degradation of Acid blue29 and Chromotrope 2R [31,32].We suppose that $CuFe_2O_4$ supported via MMT-k10 provides effective PS activation and rapid degradation of LVF.

In this study, levofloxacin (LVF), as one of extensively utilized third-generation fluroquinolones, was chosen as target antibiotic. $CuFe_2O_4$/MMT-k10 was characterized by X-ray diffraction (XRD),

X-ray photoelectron spectroscopy (XPS), scanning electron microscopy (SEM), transmission electron microscopy (TEM), the Brunner–Emmett–Teller (BET) method, and a vibrating sample magnetometer (VSM). The adsorption capacity of $CuFe_2O_4$/MMT-k10 was examined in our study. The Box–Behnken design for the response surface methodology (RSM) model was applied to optimize LVF removal in aqueous solution. Additionally, the radicals involved in the $CuFe_2O_4$/MMT-k10/PS system were revealed by free radical quenching test, as well as a potential PS activation mechanism. The purpose of this study is to explore an effective, environmentally friendly, and economical method to remove antibiotics in waste water.

## 2. Materials and Methods

### 2.1. Materials

Montmorillonite-k10 (MMT-k10) was purchased from Macklin (Shanghai, China). Ferric nitrate ($Fe(NO_3)_3·9H_2O$), cupric nitrate ($Cu(NO_3)_2·3H_2O$), citric acid monohydrate ($C_6H_8O_7·H_2O$), and potassium persulfate ($K_2S_2O_8$) were bought from Sinopharm Chemical Reagent Company (Shanghai, China). Potassium hydroxide (KOH) and sulfuric acid ($H_2SO_4$) were obtained from Guangzhou Chemical Reagent Factory (Guangzhou, China). Ethanol (EtOH) and t-butanol (TBA) from Aladdin Ltd. (Shanghai, China) were used for the quenching experiments. All chemicals were of analytical grade. All solutions were prepared with ultrapure water (resistivity > 18 MΩ).

### 2.2. Synthesis of $CuFe_2O_4$/MMT-k10 and $CuFe_2O_4$

$CuFe_2O_4$/MMT-k10 was synthesized by a citrate combustion method, according to previous research [33], with slight modification. At first, $Cu(NO_3)_2·3H_2O$ and $Fe(NO_3)_3·9H_2O$ were dissolved in 150 mL ultrapure water at a certain stoichiometric ratio and were stirred for 30 min. Then a certain amount of MMT-k10 was added into the homogeneous solution with continuous stirring for 2 h. After adding specific amount of citric acid monohydrate, the suspension was stirred vigorously at 90 °C for 2 h, and the final solid was calcined at 500 °C for 3 h. Finally, the obtained brown products were grinded to powder with mortar, washed with ultrapure water and ethanol, and dried in an oven at 55 °C for 24 h. The preparation of pure $CuFe_2O_4$ was the same method as the above-mentioned synthesis approach, but without the MMT-k10 addition.

### 2.3. Characterization of $CuFe_2O_4$/MMT-k10

The crystalline phase of $CuFe_2O_4$, MMT-k10, and $CuFe_2O_4$/MMT-k10 was analyzed by X-ray diffraction (XRD; Ultima IV, Rikagu, Tokyo, Japan) with Cu *K-α* radiation (λ = 0.15410 nm) and a 2θ range from 10° to 90°. The internal structure and external morphology of the prepared samples were observed by scanning electron microscopy (SEM; Zeiss Sigma 500) and transmission electron microscopy (TEM; FEI Tecnai G2 f20 s-twin 200kV). The Brunaure–Emmett–Teller (BET) method with pure $CuFe_2O_4$ and $CuFe_2O_4$/MMT-k10 was determined by Micromeritics ASAP 2020 from $N_2$ adsorption–desorption isotherms. A vibrating sample magnetometer (VSM; Quantum DesignSQUID, San Diego, CA, USA) was explored for measuring the magnetic properties of $CuFe_2O_4$ and $CuFe_2O_4$/MMT-k10. The surface elements of $CuFe_2O_4$/MMT-k10 before and after the reaction were measured by X-ray photoelectron spectra (XPS; Thermo Fisher Scientific K-Alpha, Boston, MA, USA).

### 2.4. Batch Experiment

The batch experiments were carried out with the addition of 0.2 g $CuFe_2O_4$/MMT-k10 into 200 mL of LVF solution (1 g/L) in conical flasks. The conical flasks were placed in a constant temperature shaker at 160 rpm under 25 °C. At certain time intervals (30 min), 5 mL of the resulting reaction solution was taken from the conical flask, filtered through a 0.45 μm membrane, and immediately measured at 285 nm [34] in a UV–1800 spectrophotometer (MAPADA, Shanghai, China). In a batch experiment, the pH of the LVF solution (pH = 6) was not adjusted. For its reusability determination, $CuFe_2O_4$/MMT-k10 was

separated after each run. After washing with ultrapure water, $CuFe_2O_4$/MMT-k10 was dried at 55 °C for the next cycle. The regenerated catalyst of heat treatment was also tested in cycle experiments. Moreover, the quench test of the scavenger was utilized for evaluation of the catalytic mechanism. In detail, 10 mL of TBA or 10 mL of EtOH was added into the reaction system before the shaker run to determine the removal rate of LVF.

*2.5. Response Surface Methodology Optimization Design*

The response surface methodology (RSM), a semi-empirical technology, is useful for designing the experiments, developing models by considering the interactions of parameters, and optimizing experimental procedures [35,36]. The Box–Behnken design has observable advantages, such as the estimation of quadratic model parameters, establishment of sequential design, investigation of the lack-of-fit of the model, and determination of block usage [37,38]. The Box–Behnken design is more efficient than the central composite design or three-level, full factorial design [39]. Hence, the Box–Behnken design was selected for this experiment. Parameters like temperature, pH value, $CuFe_2O_4$/MMT-k10 dosage, and PS dosage were considered as independent variables affecting the removal rate of LVF. As a solution pH regulator, KOH (2 mol/L) or $H_2SO_4$ (2 mol/L) were used in LVF solution. Before experiments, the pH value was recorded by a pH meter (PB-10, sartorius, Shanghai). All data was analyzed using statistical software Design Expert 8.0.

## 3. Results and Discussion

*3.1. XRD Results*

The phase structure of MMT-k10 and $CuFe_2O_4$/MMT-k10 composite was investigated by XRD pattern (Figure 1). As shown in the XRD pattern of MMT-k10, the diffraction peaks at 5.72°, 19.71°, 35.76°, 61.80°, and 73.22° were attributed to the typical crystal structure of bentonite (JCPDS No.03-0015) [40]. Moreover, the pure $CuFe_2O_4$ prepared by sol-gel method showed the diffraction peaks at 18.28°, 30.24°, 35.55°, 37.07°, 43.97°, 57.20°, and 62.43°, which was based on the analysis for the characteristic peaks' positions at the (111), (220), (311), (222), (400), (511), and (440) planes of the copper ferrite (JCPDS No.25-0283) [41]. The $CuFe_2O_4$/MMT-k10 synthetic catalysts presented here had a two-phase composition: MMT-k10 and $CuFe_2O_4$. The XRD results indicated that the $CuFe_2O_4$ particles were successfully immobilized on MMT-k10.

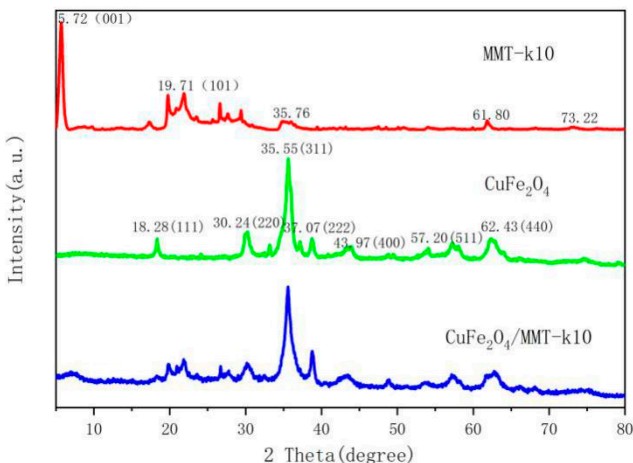

**Figure 1.** X-ray diffraction (XRD) patterns of montmorillonite k10 (MMT-k10), $CuFe_2O_4$ and $CuFe_2O_4$/MMT-k10.

### 3.2. SEM and TEM Results

The morphology and microstructure of MMT-k10, $CuFe_2O_4$, and $CuFe_2O_4$/MMT-k10 were examined using SEM/EDS and TEM. As shown in Figure 2a, the irregular lamellar structure of MMT-k10 was found. This structure was favorable for loading $CuFe_2O_4$ on the surface of MMT-k10. Generally, the sol-gel-synthesized $CuFe_2O_4$ was granular, but its magnetism makes it easy to form a large chain structure (Figure 2b). Agglomeration reduces the active site of $CuFe_2O_4$ for the $SO_4^{·-}$ AOP procedure. Interestingly, $CuFe_2O_4$ particles were seen uniformly anchored on the surface of MMT-k10 (Figure 2c). Furthermore, the EDS spectrum clearly revealed that the Cu (Figure 2d), Fe (Figure 2e), and O (Figure 2f) were uniformly distributed throughout the $CuFe_2O_4$/MMT-k10 composite, and the molar ratio of Fe/Cu was estimated to be 1.89 (close to 2.00; Figure S1). Therefore, the information of the SEM/EDS results confirmed the immobilization of $CuFe_2O_4$ on the surface of MMT-k10, which was consistent with the XRD results. In addition, TEM was used to further reveal the morphology and structure of as-prepared samples, as depicted in Figure 3. Here, as observed in Figure 3a, MMT-k10 sheets with smooth and well-defined edges were stacked together. Clearly, the $CuFe_2O_4$ particles with a size of about 10–20 nm were uniformly distributed on the MMT-k10 (Figure 3b). These results were well-suited to those from the SEM and XRD analyses.

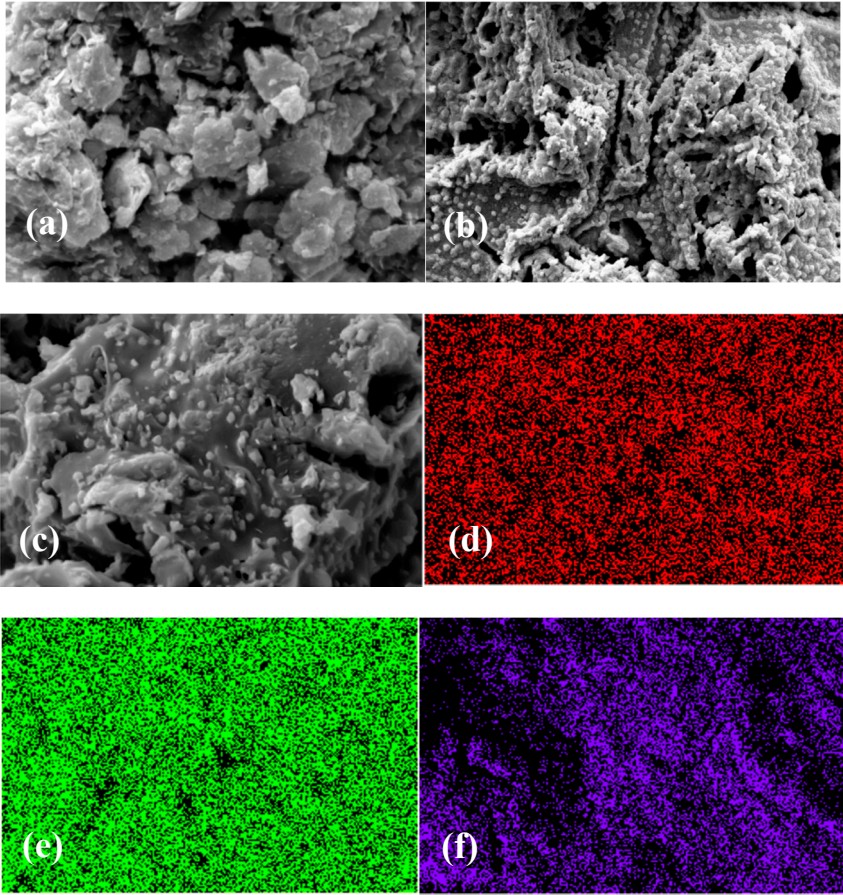

**Figure 2.** Scanning electron microscopy (SEM) images of (**a**) MMT-k10, (**b**) $CuFe_2O_4$, (**c**) $CuFe_2O_4$/MMT-k10, and EDS mapping images of Cu (**d**), Fe (**e**), and O (**f**) for $CuFe_2O_4$/MMT-k10.

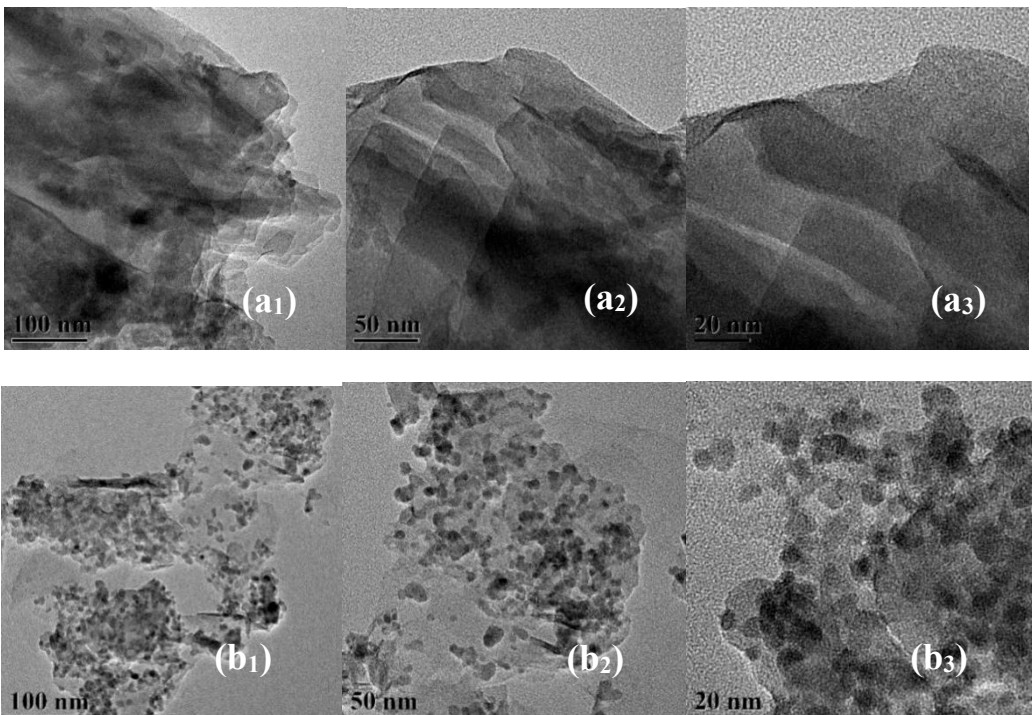

**Figure 3.** Transmission electron microscopy (TEM) images of (**a$_1$**, 100 nm; **a$_2$**, 50 nm; **a$_3$**, 20 nm) MMT-k10 and (**b$_1$**, 100 nm; **b$_2$**, 50 nm; **b$_3$**, 20 nm) CuFe$_2$O$_4$/MMT-k10.

### 3.3. BET Results

Porosity and the specific surface area of CuFe$_2$O$_4$ and CuFe$_2$O$_4$/MMT-k10 were determined by the BET technique. According to the International Union of Pure and Applied Chemistry (IUPAC) classification, the N$_2$ adsorption–desorption isotherms for both two samples (CuFe$_2$O$_4$ and CuFe$_2$O$_4$/MMT-k10) were categorized as type IV, with the apparent hysteresis loops, indicating mesoporous structures (Figure S2a). Figure S2b shows the pore size distribution of CuFe$_2$O$_4$ and CuFe$_2$O$_4$/MMT-k10 calculated by the BJH model. The pore size distributions of CuFe$_2$O$_4$ and CuFe$_2$O$_4$/MMT-k10 were in the range of 2–50 nm, with median values of 11.1 nm and 4.3 nm, respectively, which is compatible with the results of N$_2$ adsorption–desorption isotherms. However, the BET surface area of CuFe$_2$O$_4$/MMT-k10 was 66.63 m$^2$g$^{-1}$ (Table S1), much larger than that of CuFe$_2$O$_4$ (25.15 m$^2$g$^{-1}$). The larger surface area and the smaller average pore size of CuFe$_2$O$_4$/MMT-k10 than CuFe2O4 particles may make it favorable for the absorption and catalytic performance of as-prepared composites.

### 3.4. VSM Results

The magnetic property of CuFe$_2$O$_4$ and CuFe$_2$O$_4$/MMT-k10 was measured by VSM (Quantum Design SQUID) at room temperature. As can be seen in Figure 4, CuFe$_2$O$_4$ and CuFe$_2$O$_4$/MMT-k10 showed symmetrical S-type magnetization curves, indicating the super para-magnetic behavior and nanosized dimension behavior of the samples [42]. The saturation magnetization of pure CuFe$_2$O$_4$ and CuFe$_2$O$_4$/MMT-k10 were 27.1 emu·g$^{-1}$ and 13.0 emu·g$^{-1}$, respectively. The saturation magnetization value of CuFe$_2$O$_4$ was larger than that of CuFe$_2$O$_4$/MMT-k10, and this was due to the introduction of non-magnetic MMT-k10 for a hybrid catalyst. From the inset of Figure 4, CuFe$_2$O$_4$/MMT-k10 might be completely separated from the water by an external magnet.

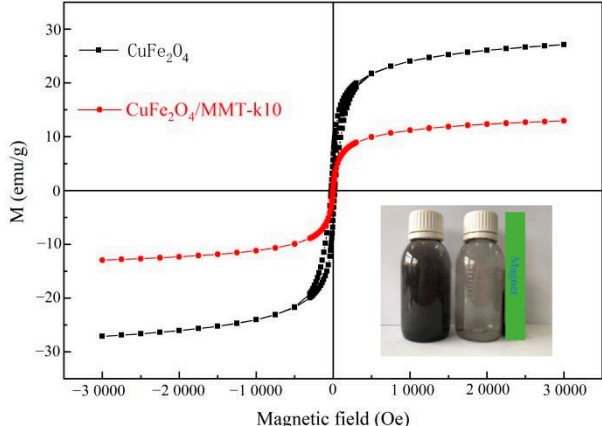

**Figure 4.** Magnetic hysteresis loops of CuFe$_2$O$_4$ and CuFe$_2$O$_4$/MMT-k10, as well as a magnetic recovery diagram (inset).

### 3.5. LVF Removal

### 3.5.1. Adsorption Studies

Adsorption equilibrium isotherms provide the most paramount parameter for designing a desirable adsorption system [43]. The isotherm for LVF adsorption on CuFe$_2$O$_4$/MMT-k10 is shown in Figure 5. Langmuir and Freundlich isotherm models were used to fit the adsorption isotherms (Figure 5). In Table S2, the results showed that it was fitted to both Langmuir ($R^2$ = 0.994) and Freundlich ($R^2$ = 0.952) isotherm models with high determination coefficients. The $R^2$ of Langmuir was higher than that of Freundlich. Hence, LVF adsorption by CuFe$_2$O$_4$/MMT-k10 tended to be monolayer adsorption. The maximum theoretical adsorption capacity of LVF was 35.8 mg/g (Table S2). Clearly, good adsorption capacity of the catalysts effectively improves the efficiency to activate PS to degrade organic pollutants [44]. The adsorption process could contribute to the removal efficiency of LVF.

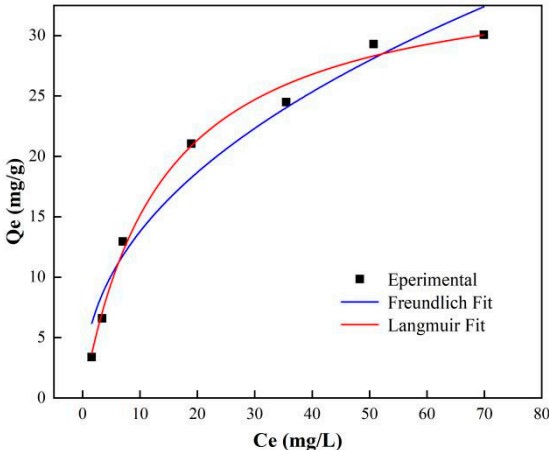

**Figure 5.** The isotherm for levofloxacin (LVF) adsorption on CuFe2O4/MMT-k10, as well as Langmuir and Freundlich isotherm model fittings.

### 3.5.2. LVF Removal in Different Systems

The LVF removal curves were displayed in different reaction systems in Figure 6. It can be seen that only 9.43% of the LVF was degraded within 120 min by PS alone; this indicates that single persulfate oxidant barely remove organic pollutant in aqueous solution [6]. Besides, LVF can hardly be effectively removed by CuFe$_2$O$_4$ and MMT-k10 (12.67% and 15.75%, respectively). Instead, 37.47% of the LVF was removed by CuFe$_2$O$_4$/MMT-k10, meaning that CuFe$_2$O$_4$/MMT-k10 had higher adsorption capacity

compared to CuFe$_2$O$_4$ and MMT-k10. Figure 5 also shows that 45.64% of the LVF was degraded by adding CuFe$_2$O$_4$ and PS, indicating that PS under the activation of CuFe$_2$O$_4$ generates many radicals for degradation. Interestingly, 85.55% of LVF was degraded in the CuFe$_2$O$_4$/MMT-k10/PS system, meaning this system was more effective than other systems in Figure 6.

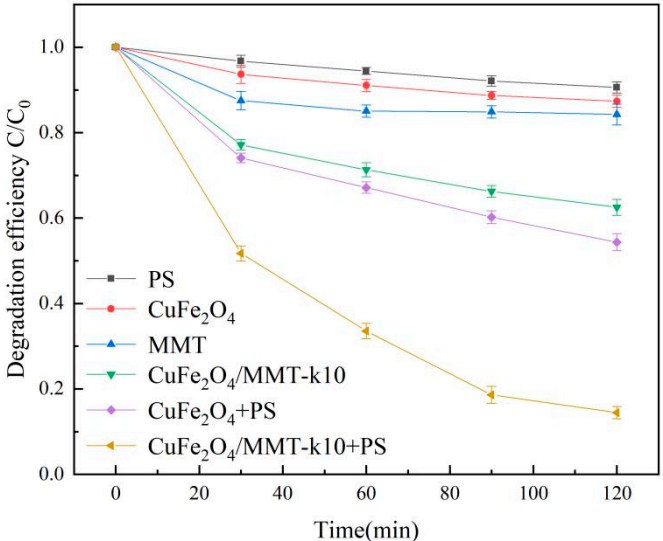

**Figure 6.** Effects of different reaction systems on the removal of LVF. Reaction conditions: $C_0$ [LVF] = 10 mg/L, $C_0$ [persulfate (PS)] = 1.0 g/L, $C_0$ [catalyst] = 1 g/L, $T$ = 25 °C and initial pH = 6 (unadjusted).

### 3.5.3. RSM Analysis of LVF Removal

In total, 29 experiments were conducted in a random order. The experimental data were analyzed by factorial regression, and the quadratic model was expressed by second-order polynomial equation.

$$Y = -20.186 + 0.185A + 1.798B + 62.587C + 61.287D + 0.021AB + 1.375AC + 2.075AD \\ - 0.170BC - 0.370BD - 3.700CD - 0.592A^2 - 0.015B^2 - 22.627C^2 - 18.827D^2$$

where $Y$ is the LVF removal rate (%), and $A$, $B$, $C$, and $D$ are independent variables that represent pH value, temperature (°C), catalyst dosage (g/L), and PS dosage (g/L), respectively. To verify accuracy of the model, a normal plot of the residuals, and a residuals vs. predicted map were adopted. As shown in Figures S3 and S4, the normal distribution of residuals was in a straight line, and the map of residuals vs. predicted was scattered, implying that the model was suitable. According to the RSM analysis, the optimal experimental conditions were compatible, as the Cu$_2$Fe$_2$O$_4$/MMT-k10 dosage was 1.5 g/L, PS dosage was 1.5 g/L, temperature was 45 °C, and pH value was 4.0. Under optimal conditions, the maximum degradation rate of LVF was 95.47%.

Analysis of variance (ANOVA) was used to test the significance and effectiveness of the second-order model, in order to predict LVF in the Cu$_2$Fe$_2$O$_4$/MMT-k10/PS system. The results are shown in Table S3. The $F$-value of the model was 21.77, and the $p$-value was less than 0.0001, indicating that the fitting result of the model was significant. In addition, the correlation coefficient $R^2$ was used to effectively evaluate the variability and applicability of the model. The correlation coefficient $R^2$ in the Cu$_2$Fe$_2$O$_4$/MMT-k10/PS system was 0.9561, representing 95.61% of the variation explained by this model. However, $R^2$ is considered a misleading statistical value, because in some cases, a high $R^2$ does not necessarily mean that the model is suitable [36]. To avoid such biases, the adjusted $R^2$ was used. Adjusted $R^2$ is a modified version of $R^2$ that is adjusted for the different number of factors in the model [36]. The adjusted $R^2$ was 0.9122 in this model, which was similar to

the correlation coefficient $R^2$. We concluded that the RSM is a suitable model to effectively determine the optimal conditions of the $Cu_2Fe_2O_4$/MMT-k10/PS system.

The correlative relationships, such as $Cu_2Fe_2O_4$/MMT-k10 dosage and PS dosage, $Cu_2Fe_2O_4$/MMT-k10 dosage and temperature, $Cu_2Fe_2O_4$/MMT-k10 dosage and pH value, and temperature and PS dosage, were evaluated. As shown in Figure 7a,b, with the increase of $Cu_2Fe_2O_4$/MMT-k10 dosage and PS dosage, the removal efficiency of LVF improved. With the increase of $Cu_2Fe_2O_4$/MMT-k10 dosage, the number of adsorption sites for LVF and the catalytic sites for PS increased. Thus, $Cu_2Fe_2O_4$/MMT-k10 dosage could improve the removal rate of LVF. PS is the main source of $SO_4^{·-}$. PS dosage increase promotes the production of $SO_4^{·-}$, which also effectively improves the removal rate of LVF. As shown in Figure 7c,d, the pH value decreases, stimulates the removal rate of LVF. To approach the actual pH value of wastewater, the pH range of 4–8 was selected. In this range, the optimum pH of $Cu_2Fe_2O_4$/MMT-k10/PS system was 4. The removal efficiency under acidic conditions is better than under alkaline conditions. This is attributed to several factors: (1) the lifetimes of HO· and $SO_4^{·-}$ decrease in alkaline conditions, but this is insufficient for the radicals to diffuse into the bulk phase for further degradation [45]; (2) under acidic conditions, $Cu_2Fe_2O_4$/MMT-k10 could release more Fe and Cu ions and activate PS to produce more $SO_4^{·-}$, in order to degrade LVF [46]. As shown in Figure 7e,f, a temperature increase promotes the removal rate of LVF. This leads to the increase of Brownian motion and the improvement of catalytic efficiency, due to thermally activated persulfate.

## 3.6. Reusability and Stability of CuFe₂O₄/MMT-k10

The stability and reusability of catalyst plays an important role in actual water treatment. Therefore, cyclic experiments were conducted to evaluate the stability and reusability of $CuFe_2O_4$/MMT-k10. Figure S5a showed the removal rate of $CuFe_2O_4$/MMT-k10 for LVF, which was reused three times. The LVF removal rate of the second and third cycles reached 72.0% and 62.43%, respectively, and $CuFe_2O_4$/MMT-k10 still maintained high efficiency. The LVF removal rate decrease may be attributed to two aspects. On the one hand, inevitable metal element consumption in a catalyst leads to the catalytic activity decrease. On the other hand, the intermediate products in the degradation process may be adsorbed on the surface of the catalyst, resulting in the reduction of absorption and catalytic activity sites. According to previous literature, the adsorbed intermediate can be removed by heat treatment, which can regenerate the catalyst [34]. Therefore, after the first cycle and the second cycle, $CuFe_2O_4$/MMT-k10 was calcined at 400 °C for 1 h. As shown in Figure S5a, after heat treatment, the catalytic performance of $CuFe_2O_4$/MMT-k10 improved significantly. Therefore, heat treatment is a kind of regeneration technology for $CuFe_2O_4$/MMT-k10, conducive to the recycling use of $CuFe_2O_4$/MMT-k10 in practical water treatment.

## 3.7. Mechanism Discussion

HO· and $SO_4^{·-}$ are the two main free radicals in the degradation of organic pollutants via PS activated by transition metal catalysts. To identify the major reactive radical species generated in the $CuFe_2O_4$/MMT-k10/PS system, free radical quenching tests were carried out. According to the previous studies, TBA reacts with HO· much more strongly than with $SO_4^{·-}$, while EtOH reacts with both HO· and $SO_4^{·-}$ at a high rate [47]. As can been seen in Figure S5b, 85.48% of the LVF was degraded in the no-scavenger system. The LVF degradation efficiency decreased to 58.98% and 30.99% while adding TBA or EtOH into the system, respectively. The results clearly reveal that both HO· and $SO_4^{·-}$ are involved in the $CuFe_2O_4$/MMT-k10/PS system, and participate in the LVF degradation process.

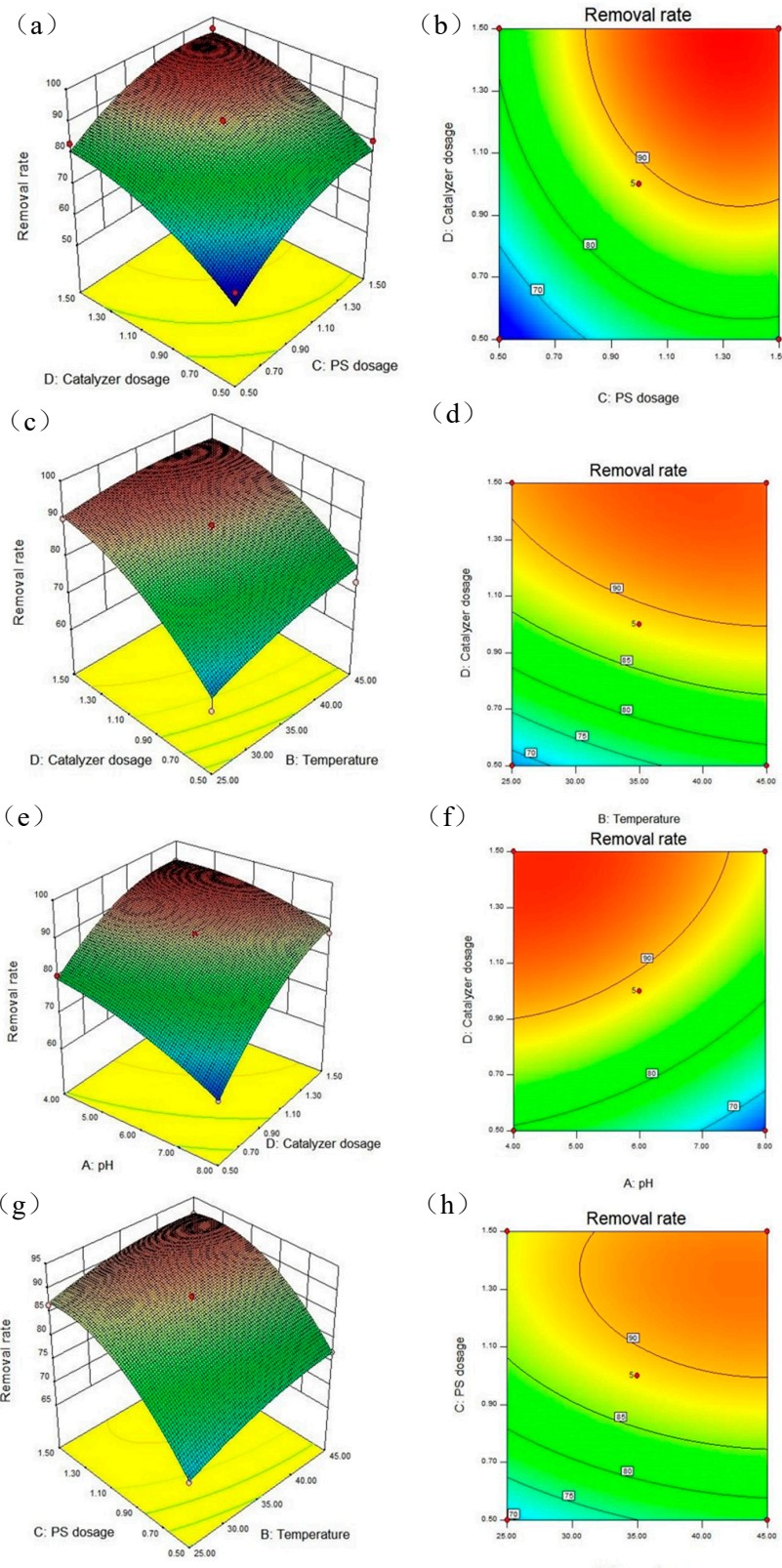

**Figure 7.** With regard to the LVF removal efficiency parameters, three-dimensional (3D) response surfaces and two-dimensional (2D) contours of Cu$_2$Fe$_2$O$_4$/MMT-k10 dosage and PS dosage (**a**,**b**), Cu$_2$Fe$_2$O$_4$/MMT-k10 dosage and temperature (**c**,**d**), Cu$_2$Fe$_2$O$_4$/MMT-k10 dosage and pH value (**e**,**f**), and temperature and PS dosage (**g**,**h**) are provided.

To better understand the surface composition and chemical states of $CuFe_2O_4$/MMT-k10 before and after use, the chemical compositions and the elemental state were characterized by XPS spectra (Figure 8). The XPS spectra of Cu 2$p$ before and after use are shown in Figure 8a. For $CuFe_2O_4$/MMT-k10 before use, the XPS peaks at 933.8 eV and 953.8 eV were assigned to Cu 2$p_{3/2}$ and Cu 2$p_{1/2}$, respectively. The splitting width between Cu 2$p_{3/2}$ and Cu 2$p_{1/2}$ was about 20.0 eV, which agreed with the standard spectrum of $Cu^{2+}$ in copper oxides, suggesting the existence of $Cu^{2+}$ in $CuFe_2O_4$/MMT-k10 [8]. In addition, it reveals that there were two obvious satellite peaks at 942.0 eV and 962.1 eV in Figure 8a, which might correspond to the open 3$d$9 shell of $Cu^{2+}$ ions [48]. Compared with the fresh $CuFe_2O_4$/MMT-k10, the new peak at 933.2 eV was formed in the hybrid catalyst after use, suggesting that the valence of Cu changed, and $Cu^{2+}$ was partially transferred to $Cu^+$ [49]. As seen from Figure 8b, the observed Fe 2$p$ 1/2 (724.4 eV) and Fe 2$p$ 3/2 (711.4eV) peaks implied that the presence of $Fe^{3+}$ and $Fe^{2+}$, respectively. At the beginning of use, $Fe^{3+}$ and $Fe^{2+}$ accounted for 63.18% and 36.82%, based on the Fe 2$p$ signal deconvolution. After the degradation experiment, no obvious peak shifting was seen in the Fe 2$p$ spectrum, but the intensity of each peak changed slightly. After the deconvolution of Fe 2$p$, $Fe^{3+}$ and $Fe^{2+}$ were detected as accounting for 70.07% and 29.93%, respectively. The results indicate that $Fe^{3+}$/$Fe^{2+}$ was involved in the $CuFe_2O_4$/MMT-k10/PS system. The O 1$s$ spectrum was revealed in Figure 8c as well. The peak at about 530.0 eV, 531.4 eV, and 532.2 eV were ascribed to surface hydroxyls (M-$(OH)_2$ or MO-OH), physic- and chemisorbed $H_2O$ on the surface, and the lattice oxygen ($O_2^-$), respectively. For fresh $CuFe_2O_4$/MMT-k10, the proportions of -OH, $H_2O$, and $O_2^-$ were 51.65%, 12.4%, and 35.87%, respectively. After use, the proportions of -OH and $H_2O$ decreased to 53.67% and 10.46%, while the proportion of $O_2^-$ was not significantly changed. The results of O 1$s$ showed that the water adsorbed on the catalyst surface was partially converted to -OH, while the composition variation was not significant. According to the above results and previous researchers, the reaction mechanism was proposed for PS activation by $CuFe_2O_4$/MMT-k10 (reactions (1–11)) [50–53], and the concept map of the proposed mechanism is shown in Figure S6. Firstly, $SO_4^{\cdot-}$ was produced by PS activated with $Cu^{2+}$ and $Fe^{2+}$ (reactions (1–2)), while HO· was produced through the $H_2O$ and $SO_4^{\cdot-}$ reaction (reaction (3)). $SO_4^{\cdot-}$ and HO· probably destroy the absorbed LVF to the degraded products (reactions (4–5)). In addition, $S_2O_8^{2-}$ possibly produces $HSO_5^-$ by means of hydrolysis (reaction (6)), and $Cu^{3+}$, $Cu^{2+}$, and $Fe^{3+}$ possibly react with $HSO_5^-$ to produce $SO_5^{\cdot-}$ (reactions (7–9)). Meanwhile, $Cu^{3+}$ likely is restored to $Cu^+$ (reaction (10)), while $Cu^+$ reacts with $Fe^{3+}$ to realize the regeneration of $Cu^{2+}$ and $Fe^{2+}$ (reaction (11)).

$$Cu(II) + S_2O_8^{2-} \rightarrow Cu(III) + SO_4^{\cdot-} + SO_4^{2-} \tag{1}$$

$$Fe(II) + S_2O_8^{2-} \rightarrow Fe(III) + SO_4^{\cdot-} + SO_4^{2-} \tag{2}$$

$$H_2O + SO_4^{\cdot-} \rightarrow H^+ + HO\cdot + SO_4^{2-} \tag{3}$$

$$SO_4^{\cdot-} + LVF \rightarrow \text{degraded products} \tag{4}$$

$$HO\cdot + LVF \rightarrow \text{degraded products} \tag{5}$$

$$S_2O_8^{2-} + H_2O \rightarrow HSO_5^- + HSO_4^- \tag{6}$$

$$Cu(III) + HSO_5^- \rightarrow Cu(II) + SO_5^{\cdot-} + H^+ \tag{7}$$

$$Cu(II) + HSO_5^- \rightarrow Cu(I) + SO_5^{\cdot-} + H^+ \tag{8}$$

$$Cu(II) + HSO_5^- \rightarrow Cu(I) + SO_5^{\cdot-} + H^+ \tag{9}$$

$$Cu(III) + H_2O \rightarrow Cu(II) + H^+ + HO\cdot \tag{10}$$

$$Fe(III) + Cu(I) \rightarrow Fe(II) + Cu(II) \tag{11}$$

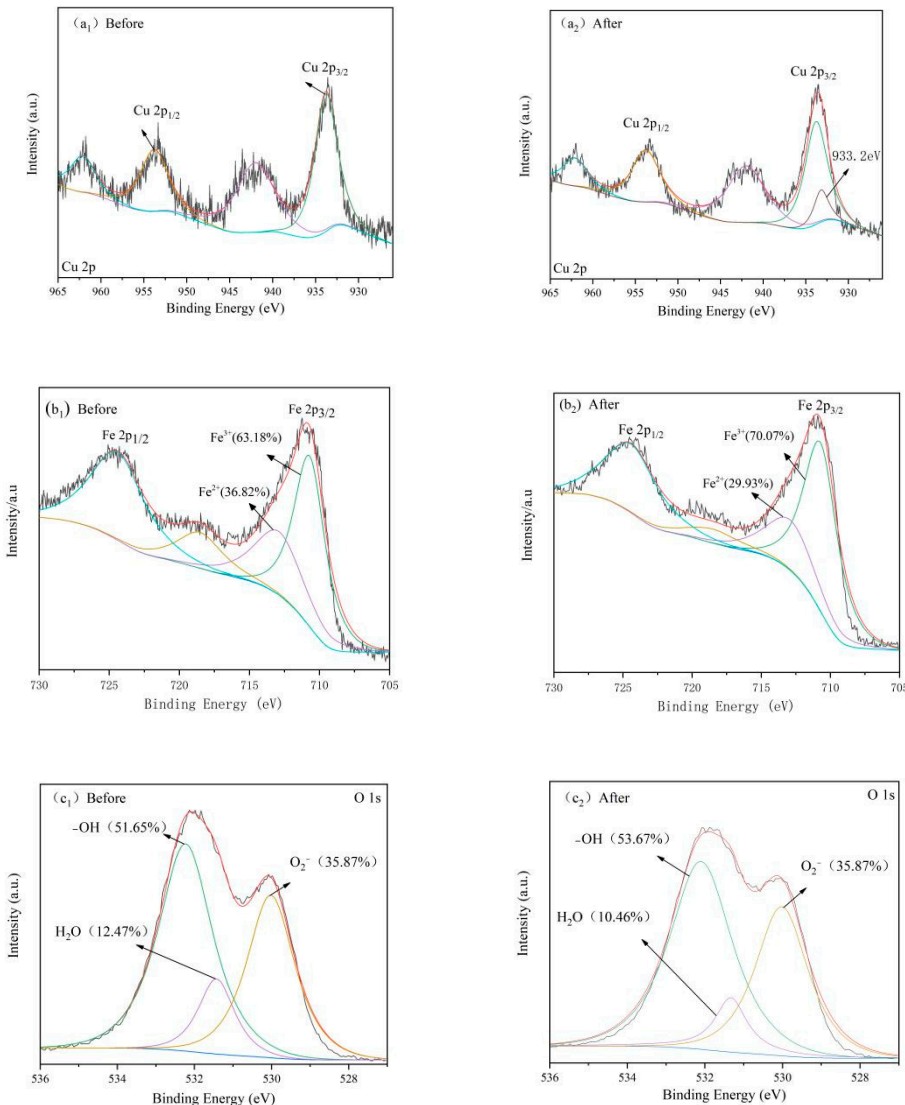

**Figure 8.** X-ray photoelectron spectroscopy (XPS) of (**a$_1$,a$_2$**) Cu 2*p*, (**b$_1$,b$_2$**) Fe 2*p*, and (**c$_1$,c$_2$**) O 1*s* enveloping Cu$_2$Fe$_2$O$_4$/MMT-k10 (1) before and (2) after use.

## 4. Conclusions

The CuFe$_2$O$_4$/MMT-k10 nanocomposite was successfully synthesized by a simple citric acid combustion method. CuFe$_2$O$_4$/MMT-k10 effectively activates PS to remove LVF in aqueous solution. In the removal process, the adsorption of LVF on CuFe$_2$O$_4$/MMT-k10 promotes this reaction. By establishing an RSM model, where the Cu$_2$Fe$_2$O$_4$/MMT-k10 dosage was 1.5 g/L, PS dosage was 1.5 g/L, temperature was 45 °C, and pH value was 4.0, the CuFe$_2$O$_4$/MMT-k10/PS system achieves the best LVF removal efficiency. The cyclic experiments confirmed the good reusability and stability of CuFe$_2$O$_4$/MMT-k10. To disclose the underlying activation mechanism, XPS was used to analyze the change of CuFe$_2$O$_4$/MMT-k10 before and after use, and the active radicals were identified by free radical quenching tests. Overall, CuFe$_2$O$_4$/MMT-k10 reveals great application potential in advanced oxidation technology for antibiotic waste water treatment.

**Supplementary Materials:** The following are available online at http://www.mdpi.com/2073-4441/12/12/3583/s1, Figure S1: EDS spectroscopy of CuFe$_2$O$_4$/MMT-k10, Table S1: Nitrogen adsorption data for CuFe$_2$O$_4$ and CuFe$_2$O$_4$/MMT-k10.

**Author Contributions:** J.Y.: Conceptualization, Data curation, Data curation, Investigation, Methodology, Project administration, Software, Visualization, Writing—original draft, Writing—review & editing.

M.H.: Supervision, Validation. S.W.: Writing—review & editing. X.M.: Conceptualization, Funding acquisition, Project administration, Resources, Writing—review & editing. Y.H.: Formal analysis. X.C.: Resources, Writing—review & editing. All authors have read and agreed to the published version of the manuscript.

**Funding:** This research was funded by [the Science and Technology Project of Guangdong Province] grant number [2017A050501029 and 2019B030301007] and [the National Natural Science Foundation of China] grant number [41071162, U1901601 and 42007084].

**Conflicts of Interest:** The authors declare that there are no conflict of interest regarding the publication of this article.

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
