# Peer review of "Efficient Removal of Levofloxacin by Activated Persulfate with Magnetic CuFe2O4/MMT-k10 Nanocomposite: Characterization, Response Surface Methodology, and Degradation Mechanism"

_water, doi:10.3390/w12123583_

Round 1

Reviewer 1 Report

The manuscript entitled “Efficient removal of levofloxacin by activated persulfate with magnetic CuFe2O4/MMT-k10 nanocomposite: Characterization, Response surface methodology, and Degradation mechanism” with manuscript ID water-1011762 by Yang et al. describes the synthesis, characterization and the use of CuFe2O4/MMT-k10 as adsorbent for levofloxacin removal from aqueous solution.

The topic is very interesting, the introduction is well written and the choice of adsorbent material is well justified.

The results are well explained; therefore I suggest the manuscript publication in Water with Minor Revisions based on the following remarks:

  1. Along the text there are some typos and confusing expression which need to be corrected, such as:  
  • “most of antibiotics were not completely transformed or dead” (lines 37-38)
  • “It seems heterogeneous catalysts is a potential method” (line 54)
  • “analytical puried” (line 103)
  • “Box-Behnken design is more efficient than the central composite design, so is the three-level full factorial design” (lines 142-143)
  • “which was belonged to the characteristic peaks” (line 153)
  • “irregular lamellar structure was found that was favorable to load CuFe2O4 on the surface of MMT-k10.” (lines 162-163)
  • “Agglomeration reduces the active site of CuFe2O4 for SO4·−-AOP in procedure” (line 165)
  • “particle may makes it favorable” (lines 194-195)
  • “both two samples” (lines 190 and 198)
  • “catalyzer” (Line 247 and Fig.6)

  1. Lines 74 and 76 a different style is used for the references: Peng el al. (2017) and Flores et al. (2020) instead of 21 and 22.
  2. In Figure 1 a bigger font should be used for the peaks numbering.
  3. If possible, I suggest changing the color or position of the scale in Fig. 2 since it is not visible.
  4. The results discussion in the lines 216-220 (adsorption studies) is confusing and repetitive since the same results are discussed in the lines 231-239 therefore I recommend deleting lines 216-220.
  5. Line 277 the authors mention the potential of zero charge (pHpzc) of Cu2Fe2O4/MMT-k10 but they don’t give its value.
  6. I suggest identification by FTIR of the intermediary products that the authors mention that may be adsorbed on the adsorption sites during the recycling experiment.
  7. The numbering of the reactions describing the mechanism should start from (1) and not (6).

I suggest the manuscript publication in Water with Minor Revisions pending the requested corrections.

Reviewer 2 Report

The novelty of the study must be better underlined. Specifically, also the findings of these three studies must be taken into consideration

  1. Salem IA, et al., Catalytic decolorization of acid blue 29 dye by H2O2 and a heterogenous

catalyst, Beni-Suef University Journal of Basic and Applied Sciences (2014), http://dx.doi.org/10.1016/j.bjbas.2014.10.003

  1. Salem IA, El-Ghamry HA, El-Ghobashy MA. Application of montmorilloniteCu(II)ethylenediamine catalyst for the decolorization of chromotrope 2R with H2O2 in aqueous solution. J Hazard Mater 2014
  2. Ibrahim A Salem, Hoda A El-Ghamry, Marwa A El-Ghobashy, Spectrochim Acta A Mol Biomol Spectrosc 2015 Mar 15;139:130-7. doi: 10.1016/j.saa.2014.11.053

The part concerning the adsorption studies is quite weak. In general, the execution of the batch tests was not adequately described. The method for the determination of residual LVF seems to be an absorption at 285 nm. Why that method was used? Are the authors sure that interferences with SO4*- are completely avoided? From Figure 5 it seems that also blank tests were carried out for comparison, but they were not adequately described. With reference to Figure 5, how many replicates were carried out? Can you add error bars?

Punctual remarks

Line 116 – is the value of the wavelength correct?

Line 127 – please detail the concentration values of LVF into the solution.

Line 128 – please detail at which time intervals

Line 131 – not clear when the pH was recorded and at which pH values the tests were carried out

Line 133 – the phase of separation must be better detailed

Line 134 – the “quench test of scavenger” was not adequately described.

I think that the results of the adsorption studies have to be moved into the main paper, conversely Table 1 can be moved into the supplementary materials, it does not bring very meaningful information.

Finally, English is not my first language, but the whole manuscript needs an accurate revision concerning language and sentence structure.

Round 2

Reviewer 2 Report

The quality of the paper has improved after revision, especially for what concerns the section describing the batch tests. The authors replied satisfactory to all comments of the reviewer.

This manuscript is a resubmission of an earlier submission. The following is a list of the peer review reports and author responses from that submission.